# Multi-hypotheses Conditioned Point Cloud Diffusion for 3D Human Reconstruction from Occluded Images

**Donghwan Kim**     **Tae-Kyun Kim**
KAIST
{kdoh2522, kimtaekyun}@kaist.ac.kr

## Abstract

3D human shape reconstruction under severe occlusion due to human-object or human-human interaction is a challenging problem. Parametric models i.e. SMPL(-X), which are based on the statistics across human shapes, can represent whole human body shapes but are limited to minimally-clothed human shapes. Implicit-function-based methods extract features from the parametric models to employ prior knowledge of human bodies and can capture geometric details such as clothing and hair. However, they often struggle to handle misaligned parametric models and inpaint occluded regions given a single RGB image. In this work, we propose a novel pipeline, MHCDIFF, **M**ulti-**h**ypotheses **C**onditioned Point Cloud **Diff**usion, composed of point cloud diffusion conditioned on probabilistic distributions for pixel-aligned detailed 3D human reconstruction under occlusion. Compared to previous implicit-function-based methods, the point cloud diffusion model can capture the global consistent features to generate the occluded regions, and the denoising process corrects the misaligned SMPL meshes. The core of MHCDIFF is extracting local features from multiple hypothesized SMPL(-X) meshes and aggregating the set of features to condition the diffusion model. In the experiments on CAPE and MultiHuman datasets, the proposed method outperforms various SOTA methods based on SMPL, implicit functions, point cloud diffusion, and their combined, under synthetic and real occlusions. Our code is publicly available at https://donghwankim0101.github.io/projects/mhcdiff.

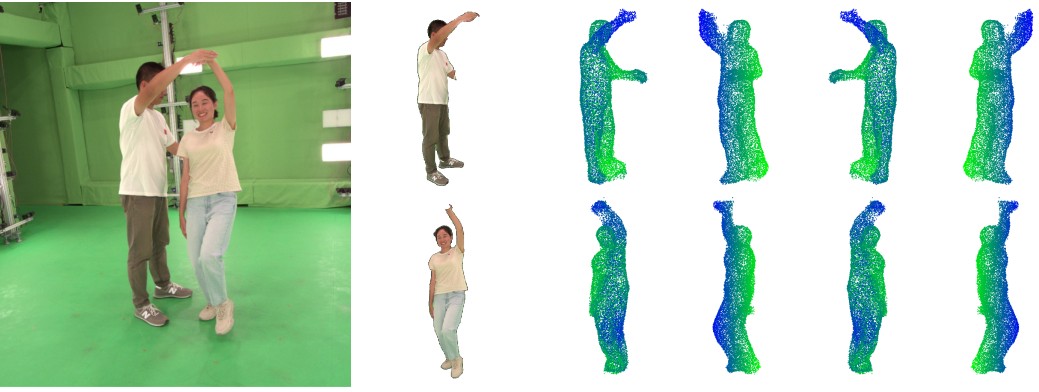

| Input image | Segmented images | 3D reconstruction as point cloud |

Figure 1: **Image to 3D shape.** From the segmented images, containing occlusion due to interaction, MHCDIFF reconstructs 3D human shapes as point clouds.

38th Conference on Neural Information Processing Systems (NeurIPS 2024).

# 1 Introduction

Realistic virtual humans play a significant role in various industries, such as metaverse, tele-presence, and game modeling. However, conventional methods require expensive artist efforts and complex scanning equipments, so they are not readily applicable. A more practical approach is to reconstruct high-fidelity 3D humans from 2D images taken in the wild. This is still an ongoing research task due to its challenges; people wear a wide variety of clothing styles and adopt diverse poses. Furthermore, human-object and human-human interaction, fundamental aspects of daily social life, make it more challenging due to severe occlusions.

Existing 3D human reconstruction methods cannot predict the *pixel-aligned* 3D shapes of humans *robustly* from occluded images. The parametric body models [27, 48, 67, 93, 75] have been widely used to reconstruct 3D human shapes. Several methods [11, 28, 34, 83, 26, 15, 10, 44, 82] predict the parameters of the statistical models and are robust to occlusion because they can be trained on large scale datasets [23, 56] and parametric models are well regularized with human body priors. However, the parametric models lack geometric details like clothing and hair, so these approaches cannot align the results to the subjects with loose clothing. More recently, 3D clothed human reconstruction methods [76, 77, 106, 92, 7, 91, 90, 95, 96], which are based on implicit functions and integrate the human body prior from the 3D body models, *i.e* SMPL [48, 67], present pixel-aligned detail shapes. Despite the impressive advances of the previous methods, they are not robust to occlusion because (1) small misalignment of estimated parametric models ruins the final shapes, (2) the implicit function takes features independently and cannot inpaint the invisible regions with missing image features, and (3) datasets [74, 87, 66] usually consists of segmented full-body images.

To address the aforementioned limitations, we propose MHCDIFF (Multi-hypotheses Conditioned Point Cloud Diffusion). (1) Several existing methods [6, 60, 73, 78, 79, 35, 8, 61, 81, 14] predict multiple SMPL meshes to model uncertainty due to occlusions. The sampled distribution is also important prior knowledge of human motions, but none of the existing work utilizes the distribution for pixel-aligned 3D human reconstruction. We leverage the multi-hypotheses to be robust on the misalignment of each sample. (2) We adopt denoising diffusion probabilistic models (DDPMs) [20] to take global consistent features and generate the invisible regions. Diffusion based methods generate 3D shapes by denoising point clouds [50, 108, 63, 22], latent [101, 62, 36], neural fields [69], 3D Gaussian [84] or meshes [46]. We adopt the unstructured point clouds to project pixel-aligned image features at each diffusion step. (3) Additionally, we synthesize partial body images by random masking [107], augmenting the limited datasets.

Specifically, our goal is *pixel-aligned* and *detailed* 3D human reconstruction in a *robust* manner to occlusion in images. Given a single occluded RGB image, we extract 2D features and generate multiple plausible SMPL hypotheses using an off-the-shelf method [4, 14]. The proposed method, MHCDIFF, Multi-hypotheses Conditioned Point Cloud Diffusion, performs the diffusion process to denoise a randomly-sampled point cloud into a target human shape. To reconstruct a pixel-aligned 3D shape and leverage the human body prior, the diffusion process is conditioned on the projected image feature (Sec. 3) and local features extracted from SMPL (Sec. 4.2). The key of MHCDIFF is a novel conditional diffusion process with multiple hypotheses (Sec. 4.3), which is not sensitive to misaligned SMPL estimation. Given global 2D features and the distribution of hypotheses, the denoising diffusion model can generate the occluded parts (Sec. 4.4)

We train MHCDIFF on randomly masked THuman2.0 dataset [85]. Our experiments on CAPE dataset [53, 68] with synthesized occlusion and MultiHuman dataset [105] with real-world interaction demonstrate that MHCDIFF reconstructs pixel-aligned 3D human shapes robustly to various occlusion ratios and achieves state-of-the-art performance. Our main contributions are as follows:

- We introduce a novel multi-hypotheses conditioning mechanism that effectively captures the distribution of multiple plausible SMPL meshes. It is robust to the noise of each SMPL estimation due to the occlusion of given images. To the best of our knowledge, MHCDIFF is the first work that extends the multi-hypotheses SMPL estimation to pixel-aligned 3D human reconstruction.

- We adopt point cloud diffusion model to capture the global consistent features and inpaint the invisible parts. Unlike the previous implicit function, the misaligned SMPL estimation can be corrected during the denoising process. The point cloud diffusion model also offers detailed human meshes.

- MHCDIFF, trained on synthesized partial body images, outperforms previous methods on occluded and even full-body images.

## 2 Related Work

### 2.1 Diffusion models for point clouds

Over the past years, denoising diffusion probabilistic models (DDPMs) [20] have been applied to point clouds. For unconditional generation, Luo et al. [50], Zhou et al. [108] and LION [101] use PointNet [70], Point-Voxel-CNN [47] and latent space, respectively. PointInfinity [22] tackles the quadratic complexity of transformer [89], and generates high-resolution point clouds with a fixed-size latent vector. Otherwise, Point-E [63] is a text-conditioned generation model using CLIP [72] and PDR [51] is a point cloud completion method from partial point clouds. PC$^2$ [57], which is the baseline of MHCDIFF, reconstructs the point cloud conditioned on projected image features (please refer to Sec. 3 for more details).

### 2.2 Explicit-shape-based human reconstruction

Parametric models [27, 48, 67, 93, 75] have been primary representations for 3D human reconstruction. Due to the strength that they capture the statistics across a large corpus of human shapes, a lot of work [11, 28, 34, 83, 26, 15, 10, 44, 82] reconstructs 3D body meshes from an RGB image. To reduce the gaps between the image and parameter space of the statistical models and improve image alignment, they propose intermediate representations or additional supervisions, such as semantic segmentation [64, 94, 33, 100] and keypoints [9, 41]. To model the uncertainty due to occlusions or depth ambiguities, some work proposes multi-hypotheses [6], heatmaps [60], probability density functions [73, 78, 79, 35] or diffusion models [8, 61, 102, 81, 39]. ProPose [14] adopts the matrix Fisher distribution [13, 30] over $\mathcal{SO}(3)$ for the joint rotation conditioned on the von Mises-Fisher distribution [55] for the unit directions of bones, which is not only mathematically correct but also learning friendly (please refer to Sec. 3 for more details). However, these methods are limited to recovering minimally-clothed humans and lack the ability to capture geometric details such as clothing and hair.

Several works aim at modeling geometric details in explicit shapes such as meshes, voxels, depth maps and point clouds. Mesh-based methods [1, 2, 3, 37, 109, 5, 25] model 3D offsets on the vertices of SMPL [48], but they do not generalize on loose clothing such as skirts and dresses. Voxel-based methods [24, 88, 17, 86] reconstruct 3D human shapes in fine-grained voxel representations. However, free-form 3D reconstruction is challenging without prior, and they need high computation costs to output high-resolution 3D shapes. Point-cloud-based methods [52, 99, 54, 19, 85] model point clouds of clothing humans. Han et al. [19] estimate depth maps based on different body parts, and convert the depth maps into point clouds. Tang et al. [85], the most related work, reconstruct 3D humans with point cloud diffusion from an RGB image. First, they convert the estimated SMPL mesh and depth map from the RGB image to point clouds. Conditioned on this point cloud, the conditional diffusion model refines the point cloud. However, they only handle complete images without occlusion and are not robust to misaligned SMPL estimation.

### 2.3 Implicit-function-based human reconstruction

Implicit-function-based methods regress occupancy fields [58] or signed distance fields (SDF) [65] utilizing Multi-Layer Perceptron (MLP) decoders as implicit functions (IF). PIFu [76] and PIFuHD [77], which are pioneering works, extract pixel-aligned image features for clothed 3D human reconstruction. Later works [106, 92, 7, 91, 90, 40, 95, 103, 104, 96] leverage parametric models or body keypoints as prior information on the human body. They extract global features from voxelized SMPL meshes with a 3D encoder [106, 90] or local features such as signed distances and normals from SMPL meshes [92, 95, 103, 104] or both [7, 96]. The use of global features helps regularize global shapes and ensure consistency and local features help reconstruct local details. However, the global encoder is sensitive to global pose changes of SMPL and decreases the performance given misaligned SMPL estimation due to occlusion. The local features do not contain the global consistent features and cannot inpaint the occluded parts. Wang et al. [90] aim to reconstruct complete 3D

shapes from occluded images by primarily using the generative global encoder with a discriminator, but only assuming the accurate SMPL meshes.

# 3 Preliminary

**PC$^2$ [57].** The projection-conditioned point cloud diffusion model is proposed for single-view 3D shape reconstruction. Denoising diffusion probabilistic model [20], which is the foundation of this framework, learns to recurrently transform noise $X_T \sim \mathcal{N}(0, \mathbf{I})$ into a sample from the target data distribution $X_0 \sim q(X_0)$ over a series of steps. In order to learn this denoising process, a neural network is trained $\mathcal{F}_\theta(X_{t-1}|X_t) \approx q(X_{t-1}|X_t)$. To reconstruct geometrically consistent 3D point clouds from single RGB images $I \in \mathbb{R}^{H \times W \times 3}$, 2D feature map $\mathcal{E}(I) \in \mathbb{R}^{h \times w \times c}$ is projected onto the partially denoised points at each step in the diffusion process. Therefore, $\mathcal{F}_\theta(\cdot) : \mathbb{R}^{(3+c)N} \to \mathbb{R}^{3N}$ is a function that predicts the noise $\epsilon \in \mathbb{R}^{3N}$ from the point cloud $X_t \in \mathbb{R}^{3N}$ and the projected features $X_t^{proj} \in \mathbb{R}^{cN}$, where $c$ is the number of feature channels.

**ProPose [14].** Recovering accurate body meshes and 3D joint rotations from single images remains a challenging problem, particularly in cases of severe occlusion, including self-occlusion and occlusion from other subjects or objects. ProPose [14] addresses this limitation by modeling the probability distributions for human mesh recovery. Since the pose parameters $\theta \in \mathbb{R}^{72}$ of SMPL [48] represent the 3D rotation of each joint and the root orientation, they adopt the matrix Fisher distribution [13, 30] over $\mathcal{SO}(3)$. Due to the gaps between the RGB images and the rotation representations, the neural network cannot easily model the distribution. ProPose [14] also introduces 3D unit vectors for bone directions as the corresponding observation on the previous matrix Fisher distribution as the prior. Leveraging Bayesian inference, they model the posterior distribution of the joint rotations from the prior distribution and observation.

# 4 MHCDIFF: Multi-hypotheses Conditioned Point Cloud Diffusion

## 4.1 Overview

Our work aims at reconstructing pixel-aligned 3D human shape as a point cloud given a single occluded RGB image via conditional point cloud diffusion, as shown in Fig. 2. Formally, the diffusion model $\mathcal{F}_\theta(\cdot)$ learns the conditional distribution $q(X_0|I)$ of 3D human shapes given the RGB images $I \in \mathbb{R}^{H \times W \times 3}$. Following PC$^2$, we extract the 2D feature map $\mathcal{E}(I) \in \mathbb{R}^{h \times w \times c}$ using ViT [12], to capture the details in the images. The image features are projected onto the partially denoised points: $X_t^{proj} = \Pi(\mathcal{E}(I), X_t)$, where $\Pi$ is the projection function. This helps obtain pixel-aligned detailed body shapes. Additionally, the diffusion model is conditioned on the local features $X_t^{SMPL}$ from SMPL mesh $S$ to exploit statistical human body priors to complete 3D shapes from occluded body parts (Sec. 4.2). However, the SMPL estimation from single occluded RGB images has a high probability of large errors. To tackle this, we propose a novel multi-hypotheses conditioned diffusion model that considers the distribution of multiple plausible SMPL meshes $\{S_i\}_{i \in \{1,...,s\}}$ (Sec. 4.3). Given the partially denoised point cloud $X_t$, the projected image features $X_t^{proj}$, and the local features from SMPL $X_t^{SMPL}$, MHCDIFF predicts the noise $\epsilon$:

$$\mathcal{F}_\theta(X_t, X_t^{proj}, X_t^{SMPL}) = \epsilon. \tag{1}$$

We also discuss how MHCDIFF takes the generative property and the global consistent features to reconstruct occluded parts (Sec. 4.4).

## 4.2 Local features from SMPL

Given the SMPL (or SMPL-X) mesh $S$ and the partially denoised point cloud $X_t$ at $t$-th diffusion step, we extract the local features $X_t^{SMPL}$ as:

$$X_t^{SMPL} = [\gamma(d(X_t|S)), \boldsymbol{n}(X_t|S)], \tag{2}$$

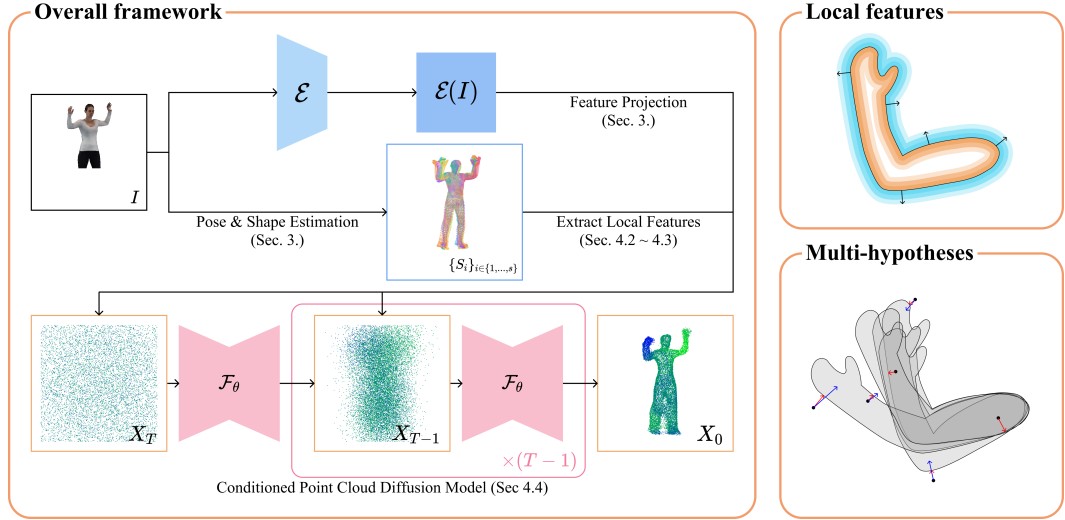

Figure 2: **(Left)** Overview of MHCDIFF. Given an occluded image $I$, MHCDIFF reconstructs 3D human shape as a point cloud. First, we extract the 2D feature map $\mathcal{E}(I)$ and hypothesize pose and shape parameters of multiple plausible SMPL meshes $\{S_i\}_{i\in\{1,...,s\}}$. Our method consists of the conditioned point cloud diffusion model (Sec. 4.4). We project the 2D image features to capture details of the image (Sec. 3) and extract local features from multiple hypothesized SMPL meshes to leverage human body priors (Sec. 4.3) **(Upper Right)** The details of local features (Sec. 4.2). The signed distance field is visualized in positive and negative regions. The arrows indicate normal vectors $\boldsymbol{n}$. **(Lower Right)** The details of multi-hypotheses (Sec. 4.3). We can consider the whole distribution during denoising process with the argmin $\bar{i}$, and the denoising can be approximated by red arrows. However, it is sensitive to extreme samples of the distribution, so we condition the mean of occupancy values, which is visualized by transparency, and the denoising can be approximated by blue arrows.

where $d(\cdot) : \mathbb{R}^3 \to \mathbb{R}$ and $\boldsymbol{n}(\cdot) : \mathbb{R}^3 \to \mathbb{R}^3$ are the signed distance and normal obtained from the closest surface of SMPL mesh respectively. In order to map scalar values to a higher dimensional space, we adopt an encoding inspired by the positional encoding in NeRF [59]:

$$\gamma(d) = (\sin(2^0\pi d), \cos(2^0\pi d), ..., \sin(s^{L-1}\pi d), \cos(s^{L-1}\pi d)). \tag{3}$$

The local features $X_t^{SMPL} \in \mathbb{R}^{(2L+3)N}$, which contain the signed distance and normal vector from SMPL, are used to predict the noise $\epsilon$ of the point cloud $X_t$. The local property, which is independent of global pose, helps MHCDIFF to generalize well in diverse SMPL estimation due to occlusion and capture local details.

### 4.3 Multi-hypotheses condition

The local features are robust to noisy SMPL estimation, but cannot correct the SMPL estimation errors. Following previous multi-hypotheses human pose estimation [6, 43, 18, 21, 8], MHCDIFF takes multi-hypotheses SMPL meshes from estimated distributions and predicts the most plausible outputs. We modify Eq. 2 to handle multiple sampled SMPL meshes $\{S_i\}_{i\in\{1,...,s\}}$ using ProPose [14] as an off-the-shelf method:

$$X_t^{SMPL} = [\gamma(d(X_t|S_{\bar{i}})), \boldsymbol{n}(X_t|S_{\bar{i}})], \tag{4}$$

where $\bar{i} = argmin_{i\in\{1,...,s\}}|d(X_t|S_i)|$, which semantically means that each point follows the closest SMPL mesh $S_{\bar{i}}$ to consider all plausible samples in denoising steps. However, each point gets conditions from only one sample and cannot leverage off-the-shelf probability distributions. In addition to the local features, we also adopt occupancy values:

$$X_t^{SMPL} = [\frac{1}{s} \sum_{i=1}^{s} \gamma(o(X_t|S_i)), \gamma(d(X_t|S_{\bar{i}})), \boldsymbol{n}(X_t|S_{\bar{i}})], \qquad (5)$$

where $o(\cdot) : \mathbb{R}^3 \rightarrow \{0, 1\}$ is the occupancy function of the given SMPL mesh, which is a binary signal while the signed distance is continuous. With the mean occupancy and closest signed distance, MHCDIFF can assume all distributions with their respective probabilities. The proposed multi-hypotheses conditioning can take an arbitrary number of SMPL, SMPL-X, and their combined.

## 4.4 Conditioned point cloud diffusion model

Finally, $\mathcal{F}_\theta(\cdot) : \mathbb{R}^{(3+c+4L+3)N} \rightarrow \mathbb{R}^{3N}$ predicts the noise $\epsilon \in \mathbb{R}^{3N}$ given the concatenation of partially denoised point cloud $X_t \in \mathbb{R}^{3N}$, projected image features $X_t^{proj} \in \mathbb{R}^{cN}$, and local features from SMPL $X_t^{SMPL} \in \mathbb{R}^{(4L+3)N}$ (Eq. 1). Notably, we do not need any learnable parameters to extract the local features from SMPL and aggregate the features of multiple SMPL meshes. We freeze the pre-trained 2D image encoder, so it is straightforward to train the diffusion model without additional training strategies.

The point cloud diffusion model of MHCDIFF takes the role of the decoder of previous implicit-function-based methods. Given the encoded features from RGB images or SMPL meshes, the decoder predicts 3D shapes such as point clouds, occupancy fields, or signed distance fields. The implicit-function-based methods need to sample the query points randomly, so the decoder has been primarily Multi-Layer Perceptron (MLP), which takes the input points independently. MHCDIFF consists of the point cloud diffusion model instead of MLP because (1) the point cloud model considers the global consistent features, (2) the diffusion model has the generative properties, and (3) the denoising process approximates correcting the misaligned SMPL estimation. Given the globally encoded image features $X_t^{proj}$ and the local features from SMPL $X_t^{SMPL}$, MHCDIFF can inpaint or restore invisible body parts and is robust to noisy SMPL estimation due to occlusion.

---

**Algorithm 1** Pseudocode of learning pipeline of MHCDIFF

---

**Require:** $\alpha_{1:T}$: diffusion noise scheduling
1: **repeat**
2:     Sample $X_0$ from $q(X_0)$
3:     Load the corresponding image $I$ and ground truth SMPL-X $S$
4:     $t \sim \text{Uniform}(\{1, ..., T\})$
5:     $\epsilon \sim \mathcal{N}(0, \mathbf{I})$
6:     $X_t = \sqrt{\bar{\alpha}_t} X_0 + \sqrt{1 - \bar{\alpha}_t} \epsilon$
7:     $X_t^{proj} = \Pi(\mathcal{E}(I), X_t)$                              ▷ Project image features (Sec. 3)
8:     $X_t^{SMPL} = [\gamma(o(X_t|S)), \gamma(d(X_t|S)), \boldsymbol{n}(X_t|S)]$
                                                       ▷ Extract local features from SMPL (Sec. 4.2
9:     Take gradient descent step on
$$\nabla_\theta \left\| \epsilon - \mathcal{F}_\theta(X_t, X_t^{proj}, X_t^{SMPL}) \right\|^2 \qquad \text{▷ Point cloud diffusion model (Sec. 4.4)}$$
10: **until** converged

---

## 5 Experiments

**Implementation.** We use the Pytorch3D library [71] for image feature projection (Sec. 3) and the kaolin library [16] to extract local features from SMPL (Sec. 4.2). MHCDIFF is trained with batch size 8 in 100,000 steps. We use MSN [4] as the image feature encoder. We use AdamW [31] with $\beta = (0.9, 0.999)$ and a learning rate which is decayed linearly from $0.0002$ to $0$. For diffusion noise schedule, we use linear scheduling from $1 \cdot 10^{-5}$ to $8 \cdot 10^{-3}$ with warmup. For inference, we denoise the point cloud for $1,000$ steps. The training process takes approximately 1 day on a single 24GB NVIDIA RTX 4090 GPU with 28M learnable parameters.

**Learning.** We synthesize the THuman2.0 dataset [98], which contains 526 high-fidelity textured scans with corresponding SMPL-X fits. We use 500 subjects for training and the others for validation.

---

**Algorithm 2** Pseudocode of inference pipeline of MHCDIFF

---

**Require:** Input image $I$
1: Sample $X_T$ from $\mathcal{N}(0, \mathbf{I})$
2: Estimate single or multi SMPL(-X) meshes $\{S_i\}_{i \in \{1,...,s\}}$
3: **for all** $t$ from $T$ to $1$ **do**
4:  $z \sim \mathcal{N}(0, \mathbf{I})$ if $t > 1$ else $z = 0$
5:  $X_t^{proj} = \Pi(\mathcal{E}(I), X_t)$           $\triangleright$ Project image features (Sec. 3)
6:  **for all** $i$ from $1$ to $s$ **do**
7:    Compute $o(X_t|S_i)$, $d(X_t|S_i)$, and $\boldsymbol{n}(X_t|S_i)$  $\triangleright$ Can be accelerated by kaolin [16]
8:  **end for**
9:  $\bar{i} \leftarrow argmin_{i \in \{1,...,s\}}|d(X_t|S_i)|$
10:  $X_t^{SMPL} = [\frac{1}{s}\sum_{i=1}^{s}\gamma(o(X_t|S_i)), \gamma(d(X_t|S_{\bar{i}})), \boldsymbol{n}(X_t|S_{\bar{i}})]$
                    $\triangleright$ Multi-hypotheses conditioning (Sec. 4.3)
11:  $\hat{\epsilon} \leftarrow \mathcal{F}_\theta(X_t, X_t^{proj}, X_t^{SMPL})$
12:  $X_{t-1} \leftarrow \frac{1}{\sqrt{\alpha_t}}(X_t - \frac{1-\alpha_t}{\sqrt{1-\bar{\alpha}_t}}\hat{\epsilon}) + \sigma_t z$    $\triangleright$ DDPM [20] sampling
13: **end for**
14: **return** $X_0$

---

We render each human subject from 36 multiple viewpoints and randomly mask the images, resulting in partially occluded body images. We use the farthest point sampling operation to sample 16,384 points from each GT scan. During the training, local features $X_t^{SMPL}$ are extracted from a single corresponding GT SMPL-X. The learning pipeline is presented in Algorithm 1.

**Inference.** First, we use the CAPE dataset [53, 68] with 150 textured scans. Similar to the training stage, we render each subject from 3 multiple viewpoints and randomly mask the images. During the inference, local features $X_t^{SMPL}$ are extracted from multiple sampled SMPL or single estimated SMPL-X. We sample 10 SMPL meshes for our experiments. To further show the generalizability on the real-world interaction, we also evaluate MHCDIFF on the MultiHuman [105] and Hi4D [97] dataset. MultiHuman, which includes the diverse interaction with objects and people, provides 3D textured scans, so we render each subject from 3 multiple viewpoints. We evaluate the performance of MHCDIFF qualitatively on Hi4D, which includes close human-human interaction with high-fidelity meshes. The inference pipeline is presented in Algorithm 2.

**Baseline models.** We compare MHCDIFF with parametric models and pixel-aligned reconstruction methods. For parametric models, which are robust for occlusion, we select ProPose [14] as SMPL estimator and PIXIE [15] as SMPL-X estimator. For pixel-aligned reconstruction methods, which can capture geometric details, we select PaMIR [106] for global features, ICON [92] for local features, and HiLo [96] and SIFU [104] for both. For the fair comparison, we primarily condition with the mean of SMPL distribution estimated via ProPose, and PIXIE is also used for ICON, which supports SMPL-X. We use pre-trained weights and evaluate under our test setting.

**Evaluation metrics.** We employ Point-to-Surface distance and Chamfer Distance as evaluation metrics. MHCDIFF outputs a point cloud, so Chamfer Distance is the average L2 distance from the reconstructed point cloud to vertices of ground-truth scans and vice versa, and Point-to-Surface distance is the average point-to-surface from the reconstructed point cloud to ground-truth scans. The outputs of implicit-function-based methods can be converted meshes via the Marching Cubes algorithm [49]. For fair comparison, we sample the same number of points from the reconstructed meshes uniformly.

## 5.1 Comparison with state-of-the-art methods

MHCDIFF outperforms prior implicit-function-based methods and SMPL estimation methods on occluded and even full-body images. Fig. 3 presents the robustness of 3D human reconstruction to the occlusion ratio. PaMIR and HiLo cannot handle the occlusions because the global feature encoder is sensitive to misaligned SMPL estimation. SIFU does not use the 3D encoder, but the cross-attention from the normal map of SMPL takes global features and is sensitive to occlusion

| | Methods | Chamfer Distance (cm) | Point-to-Surface (cm) |
|---|---|---|---|
| A | PaMIR [106] | 12.912 | 12.619 |
| | ICON [92] | 2.896 | 2.789 |
| | ICON (PIXIE estimation) | 3.329 | 3.212 |
| | SIFU [104] | 14.397 | 14.087 |
| | HiLo [96] | 13.711 | 13.405 |
| B | PIXIE (SMPL-X) [15] | 2.705 | 2.662 |
| | ProPose (SMPL) [14] | 2.370 | 2.307 |
| Ours | MHCDIFF | **1.872** | **1.810** |

Table 1: **Quantitative evaluation on CAPE dataset.** We report the average Chamfer Distance (cm) and Point-to-Surface distance (cm) on CAPE dataset. We randomly mask the images about 40% in average. We compare the performance with respect to **(A)** implicit-function-based methods; and **(B)** SMPL estimation methods used to condition MHCDIFF and (A). Best in bold, second-best underlined.

| | Methods | single | occluded single | two natural-inter | two closely-inter | three |
|---|---|---|---|---|---|---|
| A | PaMIR [106] | 0.690 | 2.349 | 5.154 | 3.752 | 4.714 |
| | ICON [92] | **0.555** | 0.549 | 0.563 | 0.786 | **0.669** |
| | SIFU [104] | 0.644 | 3.335 | 4.796 | 3.503 | 3.264 |
| | HiLo [96] | 0.606 | 2.808 | 4.139 | 3.346 | 4.398 |
| B | PIXIE (SMPL-X) [15] | 0.868 | 0.813 | 0.755 | 0.951 | 0.809 |
| | ProPose (SMPL) [14] | 0.675 | 0.567 | 0.574 | 0.766 | 0.688 |
| Ours | MHCDIFF | 0.591 | **0.491** | **0.536** | **0.703** | 0.673 |

Table 2: **Quantitative evaluation on MultiHuman dataset.** We report the average Chamfer Distance (cm) for each category. We compare the performance similar to Tab. 1.

| | | Chamfer Distance (cm) | Point-to-Surface (cm) |
|---|---|---|---|
| full | MHCDIFF | **1.872** | **1.810** |
| A | w/o occupancy | 1.893 | 1.831 |
| | w/o signed distance | 2.016 | 1.949 |
| | w/o normal | 1.888 | 1.827 |
| | w/o encoding | 1.928 | 1.863 |
| | PC$^2$ [57] | 3.640 | 3.533 |
| B | conditioned on PIXIE estimation | 2.314 | 2.237 |
| | conditioned on single ProPose estimation | 1.939 | 1.869 |
| C | trained with ProPose estimation | 2.708 | 2.624 |
| | w/o random masking | 1.940 | 1.868 |

Table 3: **Ablation study on CAPE dataset.** We validate the effectiveness of **(A)** each component; **(B)** conditioning strategies; and **(C)** training strategies.

| The number of SMPL sampled | Chamfer Distance (cm) | Point-to-Surface (cm) | Evaluation time on CAPE dataset (hours) |
|---|---|---|---|
| 1 | 1.939 | 1.869 | 4 |
| 5 | 1.882 | 1.817 | 8 |
| 10 | 1.872 | 1.810 | 12 |
| 15 | 1.833 | 1.773 | 16 |
| 20 | 1.836 | 1.777 | 20 |

Table 4: **The correlation between the number of SMPL sampled and the reconstruction quality.** We report the average Chamfer Distance (cm), Point-to-Surface distance (cm) and evaluation time of the various number of SMPL sampled.

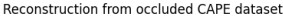

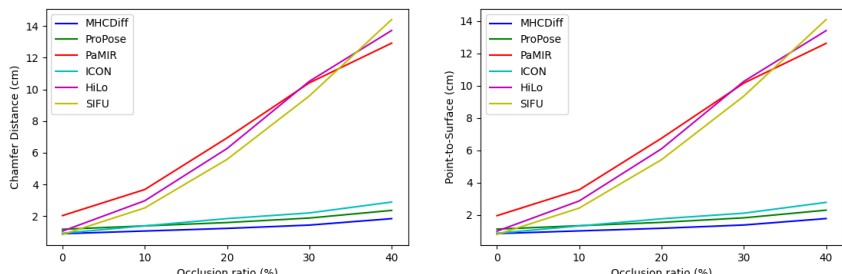

Figure 3: **A cumulative occlusion-to-reconstruction test.** This figure shows the performance of different models from the images of various occlusion ratios. From the whole-body images, which is 0% occlusion, we randomly mask the images from 10% to 40%. MHCDIFF is robust to the occlusion ratio, showing the best performance.

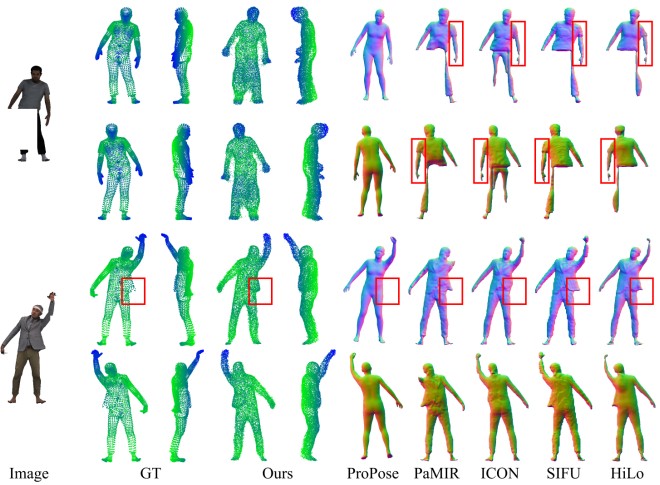

Figure 4: **Qualitative results on CAPE dataset.** We evaluate our method with SMPL estimation method and implicit-function-based methods. Given the upper image, PaMIR, ICON, and HiLo cannot generate the occluded regions. They cannot also handle the misaligned SMPL mesh on the arms, creating incomplete bodies. ProPose predicts the full-body shape, but cannot capture the details like the blazer of the lower image. However, MHCDIFF is robust to the occlusion and misalignment, and can capture pixel-aligned details.

and misaligned SMPL estimation. ICON shows comparable robustness due to its locality, but worse quality than ProPose estimation used to condition as the occlusion ratio increases. On the contrary, MHCDIFF is as robust as the statistical models, showing the most accurate results for all occlusion ratios. The results of $40\%$ occlusion ratio are also displayed in numbers in Tab. 1. Tab. 2 presents the performance on real-world interaction scenarios with MultiHuman dataset. The dataset is divided into 5 categories by the level of occlusions: "occluded single" and "two closely-inter" show the most severe occlusion, and "single" and "three" show the least occlusion. We compare the performance in each category and similar to randomly masked settings, MHCDIFF achieves state-of-the-art on severe occluded images, and comparable performance on full-body images. The major improvements of MHCDIFF are (1) correcting the misaligned SMPL estimation as shown in Tab. 1, and (2) inpainting the invisible regions as shown in Fig. 3. The qualitative results on CAPE dataset are shown in Fig. 4, and MultiHuman and Hi4D datasets are shown in the appendices Sec. E.

## 5.2   Ablation study

We conduct an ablation on MHCDIFF to validate the effectiveness of each component. In Tab. 3-B, we condition the diffusion model with single SMPL-X (PIXIE) or SMPL (ProPose) estimation.

We improve the performance with multi-hypotheses condition (Sec. 4.3). In Tab. 4, we show the correlation between the number of SMPL sampled and the reconstruction quality. More SMPL hypotheses may include more accurate samples and improve the quality (15 samples), as well as extreme samples and decrease the quality (20 samples). From PC$^2$ [57], which only takes image condition, we also validate the local features from SMPL in Tab. 3-A. All of these features improve the performance, especially the signed distance. In Tab. 3-C, MHCDIFF is trained without random masking or by conditioning the distribution estimated by ProPose [14] instead of GT SMPL-X.

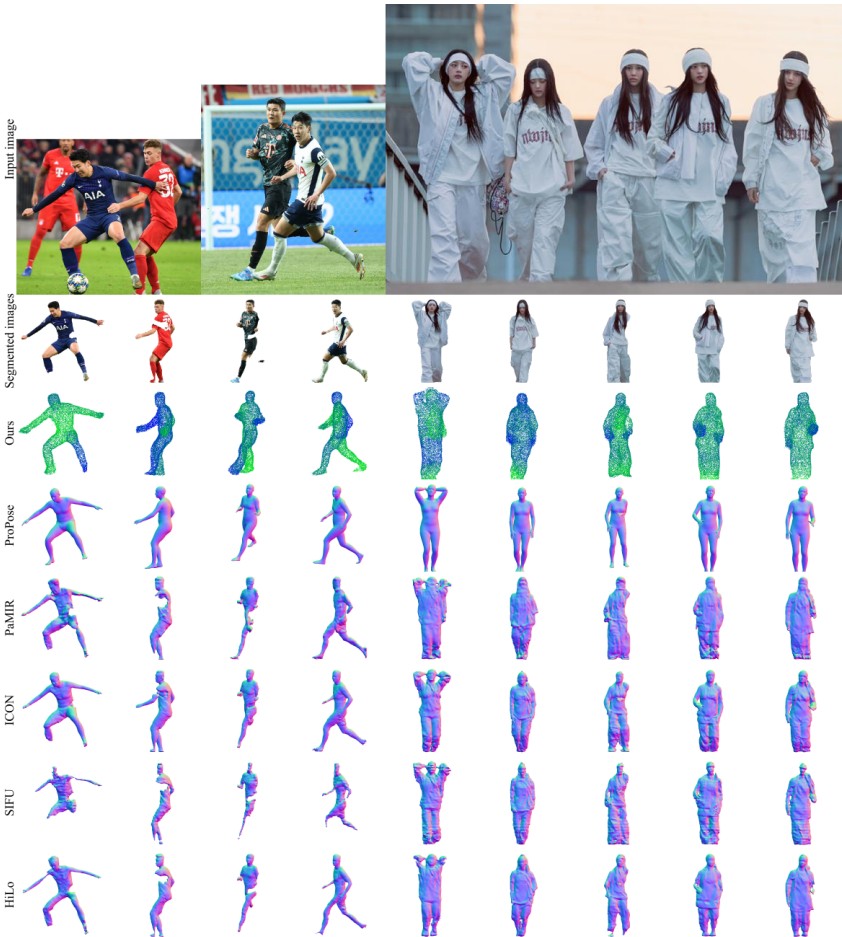

Figure 5: **Qualitative results on in-the-wild images.** Two images on the left show occlusions due to interactions, and the rightmost image shows loose clothes. From internet photos, we use [32] to segment images.

## 6 Conclusion

In this paper, We present MHCDIFF, which robustly reconstructs pixel-aligned and detailed 3D humans from single occluded images. Rather than implicit-function-based methods, we choose the point cloud diffusion model to generate invisible regions capturing the features globally. Our multi-hypotheses conditioning mechanism extracts local features from multiple SMPL estimations and integrates them without learnable parameters, so MHCDIFF is robust to a single erroneous SMPL due to occlusion. We augment the limited training data by random masking to synthesize occlusion by diverse interaction. The experiments demonstrate that our proposed method outperforms state-of-the-art methods from various levels of occlusion and interaction. In the future, the point cloud of human shapes can be applied to intermediate stages for implicit function [58] and human body deformation [45] or motion flow [38].

# 7   Acknowledgements

This work was supported by NST grant (CRC 21011, MSIT), IITP grant (RS-2023-00228996, RS-2024-00459749, MSIT) and KOCCA grant (RS-2024-00442308, MCST).

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

## A  Broader impact

Our method can be potentially used for AR/VR applications. The real-world interaction can be captured and modeled in virtual scenes, which can be extended to reinforcement learning. However, there are potential risks associated with falsifying human avatars, which could inadvertently compromise personal privacy. Consequently, there is a pressing need to establish regulations that clarify the fair use of such technology.

## B  Limitations

Our method, based on DDPM [20] sampling with $1,000$ steps, has limitation on efficiency. The training time is reasonable because we do not need query point sampling, which yields CPU bottleneck to learn implicit-function. However, evaluation on CAPE dataset takes about 12 hours, while other implicit-function-based methods take about 30 minutes. We can apply DDIM [80] sampling with fewer steps to shorten the inference time.

## C  Pointcloud to mesh

Following previous work [85, 19], we try to convert our reconstructed point cloud to mesh with Screened Poisson surface reconstruction [29]. However, the process takes about 10 hours per sample with $16,384$ points. The implicit function [58] converts the point clouds to occupancy fields by encoding features with a PointNet [70]. This two-stage pipeline can generate occluded regions and capture details. We will try this pipeline in our future work.

## D  Statistical significance

We evaluate MHCDIFF on CAPE dataset [53, 68] with 10 different random seeds. The random seeds effect on random noise in the diffusion process and SMPL sampling from the estimated distribution vis ProPose [14]. The Chamfer Distance and Point-to-Surface are $1.872(\pm0.008)$ and $1.810(\pm0.008)$ with 1-sigma error bars, respectively.

## E  Qualitative results

For the real-world interaction, we evaluate MHCDIFF on MultiHuman [105] and Hi4D [97] datasets. We render the textured scans with Pytorch3D library [71] for MultiHuman dataset, and segment each subject with pre-trained network [42] for Hi4D dataset. Our proposed method is robust not only to the occlusion but also to noise in full images or segmentation process.

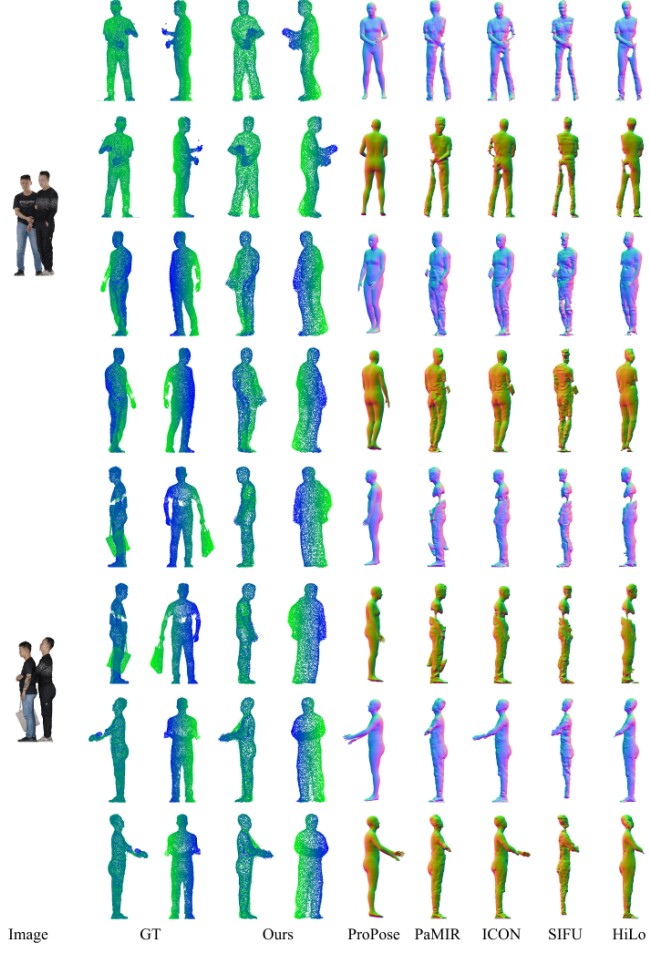

Image  GT  Ours  ProPose  PaMIR  ICON  SIFU  HiLo

Figure 6: **Qualitative results on MultiHuman dataset.**

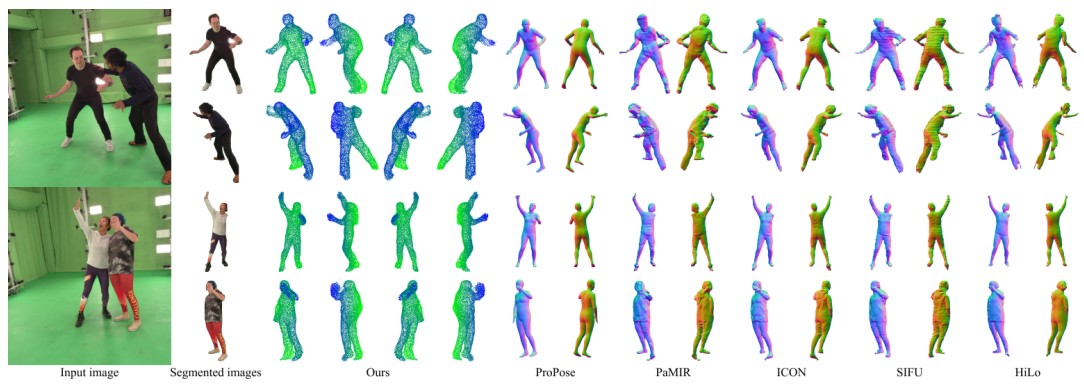

Input image  Segmented images  Ours  ProPose  PaMIR  ICON  SIFU  HiLo

Figure 7: **Qualitative results on Hi4D dataset.**

