# OpenReview forum: "Multi-hypotheses Conditioned Point Cloud Diffusion for 3D Human Reconstruction from Occluded Images"
_NeurIPS.cc/2024/Conference — NeurIPS 2024 poster_

### Official Review · Reviewer_vFH7 · 2024-07-07

**Soundness:** 3
**Presentation:** 3
**Contribution:** 3
**Rating:** 5
**Confidence:** 4

**Summary:**

Authors propose MHCDIFF (Multi-hypotheses Conditioned Point Cloud Diffusion) to reconstruct 3D human point could from a single image under occlusion. The key idea behind MHCDIFF is the smart use of image projection features and features from multiple SMPL hypothesis, coupled with point cloud diffusion.
The method achieves favourable reconstruction performance on CAPE, MultiHuman, and Hi4D datasets.

**Strengths:**

+ The proposed idea to leverage information from multiple SMPL hypothesis is simple and intuitive.
+ The ablation studies are detailed and show the significance of each component of MHCDIFF.

**Weaknesses:**

Some key details are unclear:
- L222: How are SMPL meshes sampled?
- Eq. 5: Why use mean occupancy and not mean distance? Is there an advantage of using a combination of distance and occupancy?

Experiments:
- Authors mostly evaluate their method on in door datasets with controlled settings. What about more “in the wild” data like 3DPW, MS COCO.
- Is there any correlation between the SMPL hypothesis and the reconstruction accuracy? How accurate do the hypothesis have to be? Does number of SMPL sampled matter? Does the global translation of the SMPL matter?

Minor:
- L180: shouldn’t it be argmin if we want to compute distance to closest SMPL mesh?

**Questions:**

Overall I'm positive about the work. It is interesting to see that adding basic SMPL features (average occupancy and distance to closest SMPL point) to the PC2 formulation is a useful combination. The quantitative results also look good. I'm a bit unclear on some experimental and design details as stated above.

**Limitations:**

Authors discuss limitations and broader impact in thier manuscript.

---

> ### Author Rebuttal · Authors · 2024-08-06
>
> > **Weakness 1-1.** L222: How are SMPL meshes sampled?
> >
>
> **Reply:** We sample SMPL meshes via ProPose [16] (lines 178-179). ProPose [16] predicts the distribution parameter of the matrix Fisher distribution as mentioned in Section 3 Preliminary, ProPose [16] in the paper. We sample SMPL pose parameters $\theta$ from the matrix Fisher distribution.
>
> > **Weakness 1-2.** Eq. 5: Why use mean occupancy and not mean distance? Is there an advantage of using a combination of distance and occupancy?
> >
>
> **Reply:** Distances are in wider range, so mean distance is **sensitive to extreme samples**. We use mean occupancy as **probability density function of SMPL meshes**. With the combination of distance and occupancy, MHCDIFF can assume all distributions with their respective probabilities (lines 185-186).
>
> > **Weakness 2-1.** Authors mostly evaluate their method on in door datasets with controlled settings. What about more “in the wild” data like 3DPW, MS COCO.
> >
>
> **Reply:** 3DPW and MS COCO do not have ground truth 3D shapes. Multi-view systems are necessary to capture 3D detailed scans, so there is no existing in-the-wild dataset with corresponding 3D scans. HiLo [92] and SIFU [100] use CAPE and THuman2.0 datasets, and ICON [88] use AGORA and CAPE datasets for evaluation. In Fig. R1 in PDF, We show qualitative results on in-the-wild images with occlusion and loose clothes. We will contain more results in the revision.
>
> > **Weakness 2-2.** Is there any correlation between the SMPL hypothesis and the reconstruction accuracy? How accurate do the hypothesis have to be? Does number of SMPL sampled matter? Does the global translation of the SMPL matter?
> >
>
> **Reply:** (1) The accuracy of SMPL or SMPL-X affects the reconstruction accuracy. In Tab. 1 in the paper, ProPose [16] shows better results than PIXIE [17], and the reconstruction accuracy follows the tendency in Tab. 3, group B in the paper. As shown in Fig. 4 in the paper, the accuracy of MHCDIFF follows the one of ProPose [16]. (2) In Tab. R1 in global response, we show the correlation between the number of SMPL sampled and the reconstruction quality. More SMPL hypotheses may include more accurate samples and improve the quality with 15 samples but may include extreme samples and decrease the performance with 20 samples. (3) We normalize the image with bounding box and the reconstructed results follow the global translation of SMPL estimation. In our experiment setting, sampled SMPL meshes via ProPose [16] have equal global translations. We will clarify this point in the revision.
>
> > **Weakness 3-1.** L180: shouldn’t it be argmin if we want to compute distance to closest SMPL mesh?
> >
>
> **Reply:** Thank you for pointing this out. We use the signed distance, where inside points have positive and outside points have negative values (L170), so we will change to $\bar{i}=argmin_{i\in\{1,...,s\}}|d(X_t|S_i)|$ in the revision.

---

> > ### Comment · Reviewer_vFH7 · 2024-08-14
> > **Post rebuttal update**
> >
> > Thanks authors for the rebuttal. It helped clarify my doubts.
> > I don't know why authors say AGORA dataset is not free in reply to reviewer 1vtm. To my understanding it a synthetic dataset available to download.
> > I will keep my rating to 'borderline accept'.

---

> > > ### Author Response · Authors · 2024-08-14
> > >
> > > Thank you for your valuable comment. We checked the information "AGORA is not free and public" from Tab. 1 in the paper of ICON [88], and we apologize for our mistake. We will evaluate MHCDIFF on AGORA dataset in the revision.

---

### Official Review · Reviewer_V2cx · 2024-07-08

**Soundness:** 3
**Presentation:** 3
**Contribution:** 2
**Rating:** 5
**Confidence:** 4

**Summary:**

The target of this paper is to reconstruct 3D clothed human shape. The conditional point clouds diffusion model is adopted as the main structure of the proposed reconstruction model. This work focuses on designing effective conditioning features, especially for overcoming the occlusions. The conditioning features include the similar projected image features as PC2[55], the local features [signed distance, normal vector, occupancy] extracted from multiple estimated SMPL or SMPL-X human 3D parametric models. The performance of the proposed model is evaluated on synthetic and real datasets, and compared with different SOTA methods. The experimental results show good de-occlusion ability in 3D clothed human shape reconstruction.

**Strengths:**

1.	The paper is well written and easy to understand.
2.	It is the first work that extends the multi-hypotheses SMPL estimation to pixel-aligned 3D clothed human reconstruction.
3.	The experimental results show good de-occlusion ability when reconstructing 3D clothed human 3D shape.

**Weaknesses:**

1.	Low reconstruction quality of details, such as hands and feet.
2.	The analysis and description of the experimental results are not clear and in-depth enough. For specific details, please refer to the following questions.

**Questions:**

1.	Multi-hypotheses was proposed in [8] to model the uncertainty caused by occlusions or depth ambiguities in 3D shape reconstruction. What are the differences between your work and [8] on multi-hypotheses?
2.	If one person wears different loose clothes with the same pose, can the proposed model reconstruct their 3D shapes effectively? It should be helpful if the related experimental examples are presented.
3.	Table 1 shows that the Chamfer Distances of HiLo [92] and SIFU [100] are 13.711cm and 13.397cm, while the result of the proposed method is 1.872cm, a significant improvement. My question is, both HiLo and SIFU also use SMPL model, which can provide good human 3D shape and pose prior, what is the reason for their poor results?
4.	Table 3 shows the results of ablation study. It indicates the Chamfer Distance decreases from 3.640cm (baseline PC2[55]) to 1.872cm (proposed MHCDIFF), totally 17.68mm improvement. It seems that the improvement should be contributed by the added conditioning features extracted from multiple SMPL. But from group A, the results show that each conditioning feature has small contributes, such as ‘w/o occupancy’, the value is 1.893cm vs. 1.872cm, only 0.21mm improvement. I am a bit confusing. A clearer explanation would be better.
5.	From Table 3, group B, it seems that using SMPL model (conditioned on single ProPose estimation) is better than using SMPL-X model (conditioned on PIXIE estimation). If it is true, please analyze the reasons.
6.	From the Qualitative results (Figure 3, Figure 5), it can be seen that the reconstruction qualities of some parts are not good, such as hands, feet, and so on. Please analyze the reasons.
7.	The multi-hypotheses conditioning can take an arbitrary number of SMPL, SMPL-X, and their combined. What is the optimal recommendation?

**Limitations:**

As the paper states that proposed method has low efficiency caused by diffusion model, which has limited applications.

---

> ### Author Rebuttal · Authors · 2024-08-06
>
> > **Question 1.** Multi-hypotheses was proposed in [8] to model the uncertainty caused by occlusions or depth ambiguities in 3D shape reconstruction. What are the differences between your work and [8] on multi-hypotheses?
> >
>
> **Reply:** 3D Multi-bodies [8] learns a multi-hypothesis neural network regressor to predict different SMPL hypotheses to model the uncertainty. We use ProPose [16] as multi-hypothesis SMPL estimator, which predicts the distribution parameter of the matrix Fisher distribution to solve the same problem. We can adopt 3D Multibodies [8] to estimate multi SMPL hypotheses. The main contributions of this work lie in **the use of multi hypotheses** to reconstruct full detailed body shapes.
>
> > **Question 2.** If one person wears different loose clothes with the same pose, can the proposed model reconstruct their 3D shapes effectively?
> >
>
> **Reply:** If we know different images contain one person with the same pose, we can use the same SMPL predictions, but MHCDIFF does not consider this circumstance. We show qualitative results of in-the-wild images with loose clothes. In Fig. R1 in PDF, We show qualitative results on in-the-wild images with occlusion and loose clothes.
>
> > **Question 3.** Table 1 shows that the Chamfer Distances of HiLo [92] and SIFU [100] are 13.711cm and 13.397cm, while the result of the proposed method is 1.872cm, a significant improvement. My question is, both HiLo and SIFU also use SMPL model, which can provide good human 3D shape and pose prior, what is the reason for their poor results?
> >
>
> **Reply:** HiLo [92] and SIFU [100] use implicit functions, which cannot inpaint the invisible regions (lines 34-40). Compared to ICON [88], HiLo [92] use **the global feature encoder** and SIFU [100] use **cross-attention** from the normal map of SMPL (lines 245-248). The global features are **sensitive to misaligned SMPL estimation**, so they show poor results (lines 121-123). In Fig. 3, Fig. 5 and Fig. 6 in the paper, HiLo [92] and SIFU [100] show poor qualitative results.
>
> > **Question 4.** Table 3 shows the results of ablation study. It indicates the Chamfer Distance decreases from 3.640cm (baseline PC2[55]) to 1.872cm (proposed MHCDIFF), totally 17.68mm improvement. It seems that the improvement should be contributed by the added conditioning features extracted from multiple SMPL. But from group A, the results show that each conditioning feature has small contributes, such as ‘w/o occupancy’, the value is 1.893cm vs. 1.872cm, only 0.21mm improvement. I am a bit confusing. A clearer explanation would be better.
> >
>
> **Reply:** For example, MHCDIFF ‘w/o occupancy’ is conditioned on ‘signed distance’, ‘normal’ and ‘encoding’. Among them, ‘signed distance’ is the most important component, but all of them include human 3D shape and pose prior well. However, our major contributions are **correcting the misaligned SMPL estimation** as shown in Tab. 1, group B and Tab3, group C in the paper, and **inpainting the invisible regions** as shown in Fig. 4 in the paper and Fig. R2 (Right) in PDF. We will clarify this point in the revision.
>
> > **Question 5.** From Table 3, group B, it seems that using SMPL model (conditioned on single ProPose estimation) is better than using SMPL-X model (conditioned on PIXIE estimation). If it is true, please analyze the reasons.
> >
>
> **Reply:** We select PIXIE [17] (3DV, 2021) as SMPL-X estimator following the previous pixel-aligned reconstruction methods [88, 92, 100, 102]. We could not find multi-hypothesis SMPL-X estimator, so we use ProPose [16] (CVPR, 2023). More recent SMPL-X estimators [R1, R2, R3] may show better results, but we mainly focus on multi-hypothesis estimators in this paper.
>
> > **Weakness 1.** Low reconstruction quality of details, such as hands and feet.
> >
>
> > **Question 6.** From the Qualitative results (Figure 3, Figure 5), it can be seen that the reconstruction qualities of some parts are not good, such as hands, feet, and so on. Please analyze the reasons.
> >
>
> **Reply:** We observe that the multi-hypotheses SMPL estimation via ProPose [16] shows larger variance on the hands and feet as shown in Fig. 2 (Left) in the paper. In this paper, we mainly focus on the uncertainty of bigger articulations such as arms and legs. We are currently working on this limitation and the progress is shown in Fig. R2 (Left) in PDF.
>
> > **Question 7.** The multi-hypotheses conditioning can take an arbitrary number of SMPL, SMPL-X, and their combined. What is the optimal recommendation?
> >
>
> **Reply:** As previously discussed, we focus on the uncertainty of bigger articulations, not detailed facial expressions or finger articulation. Additionally, there are few research on multi-hypotheses SMPL-X estimation, so the current optimal recommendation is using about 10 SMPL samples, as shown in Tab. R1 in global response. However, multi SMPL-X hypotheses can also be adopted without any modification.
>
> [R1] Lin et al., One-Stage 3D Whole-Body Mesh Recovery with Component Aware Transformer, CVPR 2023
>
> [R2] Cai et al., SMPLer-X: Scaling Up Expressive Human Pose and Shape Estimation, NeurIPS 2023
>
> [R3] Baradel et al., Multi-HMR: Multi-Person Whole-Body Human Mesh Recovery in a Single Shot, ECCV 2024

---

### Official Review · Reviewer_1vtm · 2024-07-13

**Soundness:** 3
**Presentation:** 3
**Contribution:** 3
**Rating:** 6
**Confidence:** 4

**Summary:**

This paper introduces method for reconstructing detailed 3D human shapes from single occluded RGB images. The key contributions are:
- A point cloud diffusion model conditioned on projected 2D image features and local features from multiple SMPL mesh hypotheses.
- A multi-hypotheses conditioning mechanism that extracts and aggregates local features from multiple plausible SMPL meshes to handle uncertainty due to occlusions.
- Training on synthetically occluded images to improve robustness to real-world occlusions.

The method is evaluated on the CAPE dataset with synthetic occlusions and MultiHuman dataset with real interactions, demonstrating state-of-the-art performance in reconstructing detailed 3D human shapes from occluded views.

**Strengths:**

- **Novel approach**: The combination of point cloud diffusion with multi-hypotheses SMPL conditioning (from ProPose) is an innovative solution to the problem of 3D human reconstruction from occluded images.
- **Robust to occlusions**: Results demonstrate significant improvement over prior work on occluded inputs.
- **Detailed reconstructions**: The approach can recover fine geometric details like clothing directly on point clouds, going beyond parametric body models.
- **Thorough evaluation**: Experiments on both synthetic (CAPE) and real (MultiHuman) datasets with varying levels of occlusion provide a comprehensive assessment.

**Weaknesses:**

- **Limited Dataset Diversity**: While results on CAPE and MultiHuman are strong, evaluation on additional datasets would further demonstrate generalization. In particular, testing on datasets with more diverse clothing styles and body shapes would be valuable. Suggested datasets include:
   - THuman2.0: Contains 2500 high-quality human scans with various poses and clothing styles.
   - AGORA: A synthetic dataset with high realism, featuring multiple people per image.
   - DeepFashion-MultiModal: Includes high-resolution images with rich annotations for clothing shapes and textures.

- **Limited practical applicability**: The method's output is a point cloud, which may not be directly usable in many real-world applications that require mesh representations. While the authors mention the possibility of converting the point cloud to a mesh using Poisson surface reconstruction, this process is computationally expensive (taking about 10 hours per sample) and is not integrated into the main pipeline. This limitation significantly reduces the method's immediate practical utility in applications requiring real-time or near-real-time 3D human reconstruction.

- **Texture and color reconstruction and real world usage**: Unlike methods like PHORHUM, this approach does not address texture or color reconstruction, which limits its applicability in photorealistic avatar creation.

**Questions:**

- How does MHCDIFF compare to recent methods like PIFuHD and PHORHUM in terms of reconstruction quality and handling of occlusions?
- Have the authors explored more efficient sampling techniques like DDIM to reduce inference time? What speedups might be achievable?
- How well does the method generalize to different clothing styles or body shapes not seen during training?
- Could the approach be extended to include texture and color reconstruction, similar to PHORHUM?
- How does the method perform on real-world, in-the-wild images with multiple humans and complex interactions? Have you considered evaluating on datasets like AGORA or DeepFashion-MultiModal?

**Limitations:**

Yes. The authors do acknowledge the computational cost as a key limitation.

---

> ### Author Rebuttal · Authors · 2024-08-06
>
> > **Weakness 1.** In particular, testing on datasets with more diverse clothing styles and body shapes would be valuable.
> >
>
> > **Question 3.** How well does the method generalize to different clothing styles or body shapes not seen during training?
> >
>
> > **Question 5.** How does the method perform on real-world, in-the-wild images with multiple humans and complex interactions?
> >
>
> **Reply:** We use THuman2.0 as training dataset. Unfortunately, AGORA is not free and public dataset and DeepFashion-MultiModal does not have ground truth 3D shapes. HiLo [92] and SIFU [100] use CAPE and THuman2.0 datasets, and ICON [88] use AGORA and CAPE datasets for evaluation. In Fig. R1 in PDF, We show qualitative results on in-the-wild images with occlusion and loose clothes. We will contain more results in the revision.
>
> > **Weakness 2.** The method's output is a point cloud, which may not be directly usable in many real-world applications that require mesh representations.
> >
>
> **Reply:** Even though point clouds may not be directly usable in real-world applications, several methods [24, 48, 49, 55, 61, 97, 104] adopt point clouds as 3d representations. Following Poisson surface reconstruction, more recent methods study surface reconstruction from point clouds [R1]. Instead of Poisson surface reconstruction, other methods or surface extraction using the marching cube [47] proposed in PointInfinity [24] may be the solution, but these are not our main scope. We will consider this aspect in our future work.
>
> > **Weakness 3.** Unlike methods like PHORHUM, this approach does not address texture or color reconstruction, which limits its applicability in photorealistic avatar creation.
> >
>
> > **Question 4.** Could the approach be extended to include texture and color reconstruction, similar to PHORHUM?
> >
>
> **Reply:** We use PC2 [55] as the baseline of MHCDIFF, and they predict the color of each point using separate models with equivalent architecture. Therefore, we can extend to include color reconstruction, but we only focus on shape reconstruction in this paper.
>
> > **Question 1.** How does MHCDIFF compare to recent methods like PIFuHD and PHORHUM in terms of reconstruction quality and handling of occlusions?
> >
>
> **Reply:** Similar to pixel-aligned reconstruction methods, PaMIR [102], ICON [88], HiLo [92], SIFU [100] (lines 231-232), PIFuHD and PHORHUM use implicit functions, which cannot inpaint the invisible regions (lines 34-40). Including PIFuHD and PHORHUM, they can capture geometric details of visible parts, but cannot handle occlusions at all.
>
> > **Question 2.** Have the authors explored more efficient sampling techniques like DDIM to reduce inference time? What speedups might be achievable?
> >
>
> **Reply:** We have explored DDIM to reduce inference time, but the model cannot fully denoise and produce blurred point cloud results. We can use lower-resolution point clouds to reduce inference time while losing some details.
>
> [R1] Huang et al., Surface Reconstruction from Point Clouds: A Survey and a Benchmark, ArXiv 2022

---

### Official Review · Reviewer_w3VR · 2024-07-13

**Soundness:** 2
**Presentation:** 1
**Contribution:** 2
**Rating:** 3
**Confidence:** 4

**Summary:**

This paper investigates the problem of improving the robustness of occluded 3D human reconstruction from a single image. The idea is to achieve better stability under occlusion via two steps: 1. refining pixel-aligned local image feature extraction part with the help of multi-hypothesis human pose and shape estimation; 2. inpainting the invisible body part via a point cloud diffusion process. Experiments on CAPE and MultiHuman datasets show the superior of the proposed method.

**Strengths:**

1. The ablation study in Tab. 3 reveals the effectiveness of each designs.
2. Using point cloud diffusion process to inpainting the invisible part seems reasonable.

**Weaknesses:**

1. Composing the multi-hypothesis human pose and shap with point cloud diffusion seems incremental. The paper lacks solid insight to inspire the readers.
2. The quantitative results on CAPE and MultiHuman don't show obvious improvement, compared with previous methods. The CAPE dataset is processed to "randomly mask the images about 40% in average". While on MultiHuman without any masking, the performance is quite similar to the baselone model.
3. More qualitative results would be helpful to determine the superior of the proposed method.

**Questions:**

What is the key insight behind the two designs presented in this paper?
The experiment results on CAPE and MultiHuman may need further explanation.

**Limitations:**

No.

---

> ### Author Rebuttal · Authors · 2024-08-06
>
> > **Weakness 1.** Composing the multi-hypothesis human pose and shape with point cloud diffusion seems incremental. The paper lacks solid insight to inspire the readers.
> >
>
> > **Question 1-1.** What is the key insight behind the two designs presented in this paper?
> >
>
> **Reply:** We design MHCDIFF with the two steps with the insight in Section 1 Introduction (lines 41-51). (1) We adopt multi-hypothesis human pose and shape to model uncertainty (lines 43), and be **robust on the misalignment due to occlusion** (lines 45-46). (2) We choose the point cloud diffusion model to **generate the invisible regions** by taking global consistent features (lines 46-47). Among 3d representations, point clouds are effective to project pixel-aligned image features at each diffusion step (lines 49-50).
>
> > **Weakness 2.** The quantitative results on CAPE and MultiHuman don't show obvious improvement, compared with previous methods. The CAPE dataset is processed to "randomly mask the images about 40% in average". While on MultiHuman without any masking, the performance is quite similar to the baseline model.
> >
>
> > **Question 1-2.** The experiment results on CAPE and MultiHuman may need further explanation.
> >
>
> **Reply:** MultiHuman dataset is divided into 5 categories by the level of occlusions and the number of people. In Tab. 2 in the paper, “occluded single” and “two closely-inter” show the most severe occlusion, and “single” and “three” show the least occlusion. For “occluded single” and “two closely-inter”, MHCDIFF achieves state-of-the-art performance, and the results have a similar tendency to CAPE dataset with 10~20% occlusion ratios in Fig. 4 in the paper. The major improvement of MHCDIFF is **correcting the misaligned SMPL estimation** as shown in Tab. 1, group B and Tab3, group C in the paper, and **inpainting the invisible regions** as shown in Fig. 4 in the paper and Fig. R2 (Right) in PDF. We will clarify this point in the revision.
>
> > **Weakness 3.** More qualitative results would be helpful to determine the superior of the proposed method.
> >
>
> **Reply:** In Fig. R1 in PDF, We show qualitative results on in-the-wild images with occlusion and loose clothes. We will contain more results in the revision.

---

### Author Rebuttal · Authors · 2024-08-05

We thank the reviewers for their thoughtful feedback and finding that the proposed method is “simple and intuitive” [vFH7] and shows “significant improvement” [1vtm, V2cx]. We also appreciate [V2cx] from finding that “paper is well written and easy to understand.”

|The number of SMPL sampled|CD (cm)|P2S (cm)|Evaluation time on CAPE dataset (hours)
|---|---|---|---|
|1|1.939|1.869|4|
|5|1.882|1.817|8|
|10|1.872|1.810|12|
|15|1.833|1.773|16|
|20|1.836|1.777|20|
[Table R1: The correlation between the number of SMPL sampled and the reconstruction quality. CD and P2s denote Chamfer Distance and Point-to-Surface distance, respectively.]

 We provide the figures referred to in our author response in the PDF file below.

---

### Decision · Program_Chairs · 2024-09-25

**Decision:**

Accept (poster)

**Comment:**

3 out of 4 reviewers recommended acceptance. the reviewer who recommended reject is mainly concerned about the main ideas: Multi-hypotheses poses and diffusion-based framework, are incremental. These are viewed by other reviewers as simple, effective and beneficial. This AC follows the majority views on this paper to be accepted.